# Exploring the Mechanism of Gyejibokryeong-hwan against Atherosclerosis Using Network Pharmacology and Molecular Docking

**DOI:** 10.3390/plants9121750

**Published:** 2020-12-10

**Authors:** A Yeong Lee, Joo-Youn Lee, Jin Mi Chun

**Affiliations:** 1Herbal Medicine Resources Research Center, Korea Institute of Oriental Medicine, Naju-si 58245, Korea; lay7709@kiom.re.kr; 2Therapeutics and Biotechnology Division, Korea Research Institute of Chemical Technology, 141 Gajeong-ro, Yuseong-gu, Daejeon 34114, Korea; leejy@krict.re.kr

**Keywords:** herbal formula, Gyejibokryeong-hwan, atherosclerosis, network pharmacology, molecular docking

## Abstract

Gyejibokryeong-hwan (GBH) is a traditional formula comprised of five herbal medicines that is frequently used to treat blood stasis and related complex multifactorial disorders such as atherosclerosis. The present study used network pharmacology and molecular docking simulations to clarify the effect and mechanism of the components of GBH. Active compounds were selected using Oriental Medicine Advanced Searching Integrated System (OASIS) and the Traditional Chinese Medicine System Pharmacology Database and Analysis Platform (TCMSP), and target genes linked to the selected components were retrieved using Search Tool for Interacting Chemicals (STITCH) and GeneCards. Functional analysis of potential target genes was performed through the Annotation, Visualization and Integrated Discovery (DAVID) database and the Kyoto Encyclopedia of Genes and Genomes (KEGG) pathway, and molecular docking confirmed the correlation between five core compounds (quercetin, kaempferol, baicalein, ellagic acid, and baicalin) and six potential target genes (AKT1, CASP3, MAPK1, MAPK3, NOS2, and PTGS2). Molecular docking studies indicated that quercetin strongly interacted with six potential target proteins. Thus, these potential target proteins were closely related to TNF, HIF-1, FoxO, and PI3K-Akt signal pathways, suggesting that these factors and pathways may mediate the beneficial effects of GBH on atherosclerosis. Our results identify target genes and pathways that may mediate the clinical effects of the compounds contained within GBH on atherosclerosis.

## 1. Introduction

In traditional Korean medicine, blood stasis is an important pathological feature of various disorders, including hyperviscosity, hyperlipidemia, inflammation, ischemic brain injury, and atherosclerosis [1,2]. For example, there is an association between blood stasis syndrome score and the atherosclerosis index in stroke patients [3]. Atherosclerosis is a chronic complex, and multifactorial inflammatory disease characterized by lipid accumulation, smooth muscle cell proliferation, cell apoptosis, necrosis, fibrosis, and local inflammation [4]. There are several risk factors for atherosclerosis, including abnormal lipid metabolism, inflammatory reactions, and dysfunctional arterial endothelial cells, and their involvement in disease pathogenesis is complicated [5]. The effects and mechanisms of various herbal-based mixtures to prevent and treat atherosclerosis have been explored in many studies [6,7,8,9].

Gyejibokryeong-hwan (GBH; known as Guizhi-fuling-wan in China and Keishibukuryo-gan in Japan) is a traditional herbal formula consisting of five herbal medicines (*Cinnamomum cassia* J. Presl (Lauraceae), *Poria cocos* Wolf (Polyporaceae), *Paeonia suffruticosa* Andrews (Paeoniaceae), *Paeonia lactiflora* Pallas (Paeoniaceae), and *Prunus persica* Batsch (Rosaceae)) [10]. GBH has been extensively used throughout Asia and is approved by the Korean Ministry of Food and Drug Safety and the US Food and Drug Administration [10,11]. GBH is used to treat menopausal symptoms and blood stasis [2,12,13,14], and it reportedly protects against atherosclerosis [15], inhibits platelet aggregation [16], displays anti-hypercholesterolemia activity [17], and improves blood circulation [18]. Although GBH is frequently used to treat bloodstasis-related atherosclerosis in Korea, it is difficult to determine its mechanism of action because it contains a large number of active ingredients. Traditional Korean and Chinese medicine uses herbal formulas to treat disease and is based on the theories of a holistic approach to treatment and symptom pattern differentiation [19]. Thus, studies that investigate the mechanisms and multiple targets that underlie herbal formulas containing numerous active ingredients are needed. The identification of gene networks regulated by active ingredients is one way to garner information about how ingredients achieve their specific therapeutic efficacy. Recently, network pharmacology has been used to better understand the mechanism of action of herbal medicines. Network pharmacology analysis can improve the effective use of herbal medicines by predicting the potential pharmacological targets for the active ingredients in candidate herbal medicines [20]. This approach seeks to identify the key nodes within disease-relevant networks whose simultaneous targeting results in the system-wide disruption that is needed to treat complex diseases [21]. To quickly validate the correlations between active ingredients and potential target genes from network pharmacology analysis, many researchers have used molecular docking methodology [22,23]. Molecular docking is a modeling technique that maps small ligands into macromolecular target structures to identify potential molecular interactions at the binding site; the technique generates a so-called docking sore that is widely used for hit identification during computer-added drug design [24,25].

This study used network pharmacology to explore the underlying mechanisms and targets of GBH for treating bloodstasis-related atherosclerosis. In addition, molecular docking was used to validate correlations between the active ingredients and potential target genes identified by network pharmacology. Using these methods, we developed a network target prediction, based on herb-compound-target and gene-pathway interactions, to evaluate the relationships between key active ingredients in GBH and its pharmacological mechanisms.

## 2. Results

### 2.1. Components of GBH

To identify the components of GBH, we performed gas chromatography mass spectrometry (GC-MS; Figure 1A) and liquid quadrupole mass detector (LC-MS; Figure 1B) pattern analysis for methanol extracts of GBH. Fourteen compounds were detected using GC-MS and comparing the National Institute Standards and Technology Mass Spectral (NIST MS) library, and nine compounds were confirmed using LC-MS and comparing standard compounds. A total of 21 compounds were confirmed (two compounds (cinnamaldehyde and paeonol) were duplicated): benzoic acid, (Z)-cinnamaldehyde, (E)-cinnamaldehyde, (+)-cyclosativene, α-copaene, sucrose, paeonol, Isopropyl-4,7-dimethyl-1,2,4a,5,6,8a-hexahydronaphthalene, α-muurolol, (+)-cadinene, 4-isopropyl-1,6-dimethyl-1,2,3,4,4a,7-hexahydronaphthalene, tau-muurolol, tans-oleic acid, 11-oxatetacyclo (5.3.2.0(2,7).0(2,8)dodecane-9one, amygdalin, paeonolide, albiflorin, paeoniflorin, ellagic acid, bezoylpaeoniflorin, trans-cinnamic acid, paeonol, cinnamaldehyde. The GC-MS and LC-MS profiles of GBH are shown in Figure 1.

Through searching the Traditional Chinese Medicine System Pharmacology Database and Analysis Platform (TCMSP), we identified 422 compounds in GBH, including compounds present in bark of *Cinnamomum cassia* J. Peresl (*n* = 206), sclerotium of *Poria cocos* Wolf (*n* = 33), root of *Paeonia suffruticosa* Andrews (*n =* 54), root of *Paeonia lactiflora* Pallas (*n* = 114), and seed of *Prunus persica* Batsch (*n* = 65). Of the 422 compounds, 50 were duplicated. In addition, 29 compounds were identified from searches of published chemical profiling reports [10,26] and 21 compounds were detected through LC-MS and GC-MS profiling. A total of 437 compounds were found through databases, previous reports, and chemical profiling analysis using LC-MS and GC-MS; 35 were duplicated (Appendix A).

### 2.2. Absorption, Distribution, Metabolism, and Excretion (ADME) Screening of Selected GBH Components

In this study, threshold values for our ADME evaluation system were oral bioavailability (OB) ≥ 30% and drug-likeness (DL) ≥ 0.18, as recommended by TCMSP guidelines [7]. Even though three compounds (paeonol, amygdalin, and cinnamaldehyde) had values lower than the ADME criteria, they were included because they are the main active compounds in *Paeonia suffruticosa* Andrews, *Prunus persica* Batsch, and *Cinnamomum cassia* J. Presl, respectively, according to the Korean Pharmacopoeia criteria [7,20]. These compounds are not only the main active compounds in GBH, but are also involved in the anti-atherosclerosis effects [27,28,29,30,31]. Finally, 74 compounds passed the ADME screening process (Figure 2).

### 2.3. Relationships between Target Compounds and Genes

We searched 4700 genes linked to 35 compounds (Appendix A) and selected 273 genes with confidence scores ≥ 0.700. The computing method is described in the literature, and genes with a confidence score ≥ 0.700 were considered high-confidence compounds [32,33]. In general, a higher confidence score indicates that the relationship between a target compound and a gene is more likely to be accurate [33]. A total of 20 compounds and 273 genes were selected as a result of conducting network analysis without weights, consisting of 293 nodes and 412 edges (Table 1). The topological properties of every node in the interaction network were analyzed, and the results were ranked by degree of node, closeness centrality, and betweenness centrality. Nodes (targets) with higher ranks were considered to have a more critical role within the network. As a result, quercetin was the most important node among 20 compounds in this network because of the high degree of node, closeness centrality, and betweenness centrality. β-Sitosterol α1, which was only linked to genes (MSMO1 and C5orf4), did not network with other compounds despite its high closeness centrality and betweenness centrality. Also, dehydroeburicoic acid and amygdalin were related with one gene, so they were not connected in this network. The degree of node is the number of edges connected to it [34]. Betweenness centrality determines the frequency with which shortest paths between any pair of nodes pass through that node [35]. Closeness centrality is defined as the reciprocal of the average of the shortest path distance between a node and all other nodes in the network; the larger this value is, the greater the centrality of the node is, indicating that the signal is passed from one node to other nodes faster, A larger value for degree of node, betweenness centrality, and closeness centrality in the network indicates that the node has greater importance [36].

### 2.4. Network Analysis of Potential Compounds and Target Genes

Target genes associated with atherosclerosis were searched against the GeneCards database, which identified genes (*n* = 4346) related to atherosclerosis (Appendix A). Following matching of the aforementioned 273 target genes, 215 genes overlapped with atherosclerosis-related genes. In this network, 18 compounds and 215 atherosclerosis-related genes formed 233 nodes and 345 edges and allowed us to compile a list of target genes that interacted with the compounds of GBH. As a result of topological analysis of this network, the average number of neighbors was 2.978; only 46 nodes had above the average number of neighbors. Among the 233 nodes, quercetin had the highest degree of node, betweenness centrality, and closeness centrality. The top five nodes of this network were quercetin, baicalein, kaempferol, ellagic acid, and baicalin, according to the degree of node, closeness centrality, and betweenness centrality. As a result of topology analysis for genes excluding compounds, the top 6nodes for genes were: CASP3, NOS2, PTGS2, MAPK3, MAPK1, and AKT1 according to the degree of node; MSMO1, NOS2, MMP2, MAPK3, MAPK1, and AKT1 according to high value of closeness centrality (however MSMO1 was excluded because it did not connect a network); AKT1, CAPS3, ICAM1, NOS2, TNF, and MMP2 according to high value of betweenness centrality (however ICAM1, TNF, and MMP2 were low degree of node). Overall, there were six genes, AKT1, CAPS3, MAPK1, MAPK3, NOS2, and PTGS2, with high values of degree of node, closeness centrality, and betweenness centrality. Quercetin was the most important node in this network because of the highest degree of node, betweenness centrality, and closeness centrality. In addition, this compound was associated with the most atherosclerosis-related genes with the highest confidence score (Appendix A). We suggested that these targets may be key nodes responsible for the protective effects of GBH against atherosclerosis (Figure 3).

In order to determine the extent to which the five medicines contribute to the treatment of atherosclerosis, a network analysis performed between active compounds derived from herbs and atherosclerosis–related genes. Further linking of target compounds with atherosclerosis–related genes resulted in a network of five herbs, 18 compounds, and 215 target genes (Figure 4A). We represented the contribution of herbs containing active compounds as a Sankey diagram based on this network. In the herbs-compounds-genes network, atherosclerosis-related genes were closely related to the active compounds in *Paeonia suffruticosa* Andrews (Figure 4B).

### 2.5. Pathway Analysis of Potential Targets of GBH

To investigate the mechanisms underlying the atherosclerosis-related effects of GBH, we performed functional enrichment analysis using the Kyoto Encyclopedia of Genes and Genomes (KEGG) pathway database. We selected targets (*n* = 74) that were associated with more than two compounds from potential target genes (*n* = 215). The results of KEGG pathway analysis (*p*-value < 0.05) identified various functional, signal transduction, endocrine system, and other pathways potentially related to atherosclerosis (Figure 5).

Target genes were closely related to several signal transduction-related pathways, including tumor necrosis factor (TNF), HIF-1, FoxO, and PI3K-Akt signaling pathways (Figure 6A). The contribution of the active GBH compounds to the atherosclerosis-related KEGG pathways is shown in Figure 6B; the three compounds (30% of quercetin, 19% of baicalein, and 13% of kaempferol) had the most frequently acted on KEGG pathways linked to atherosclerosis. These results suggest that GBH modulates several atherosclerosis-related signaling pathways and genes.

### 2.6. Molecular Docking Studies

Five active compounds and six potential target genes were selected through network analysis with high node and confidence scores. Molecular docking studies were carried out to confirm the interaction between active compounds and atherosclerosis-related potential target genes at the molecular level. Molecular docking using the X-ray crystal structures of six target proteins identified by network analysis was used to predict the potential binding models of five compounds of GBH. The docking scores between five compounds and six target proteins are represented in Table 2. In molecular docking simulation, the docking score is a tool to estimate the binding energy of a ligand, and a lower negative value means that it has a strong binding force.

Baicalein, baicalin, and quercetin bound to the ATP binding pocket of the kinase domain in AKT1, MAPK1, and MAPK3. The hydroxy group of flavonoids made hydrogen bonds with the hinge region. In addition, the aromatic ring bound to the hydrophobic pocket. In CASP3, four compounds (baicalin, ellagic acid, kaempferol, and quercetin) bound to Cys163 residue. These compounds all showed common hydrophilic/hydrophobic interactions with Cys163 and His121 in their predicted binding poses. In addition, baicalein, baicalin, kaempferol, and quercetin were attached to NOS2. These compounds placed the aromatic ring group almost parallel to the heme in the binding site, which was defined by Glu377 and Pro350 in the binding cavity. Because the aromatic ring groups of the compounds lay almost parallel to the heme group, a series of π–π interactions were observed.

Baicalein and quercetin bound to the active site in PTGS2; the hydroxyl group of baicalein and quercetin formed hydrogen bonds with the side chain of catalytic Tyr385, located at the top of the active site. Also, 4-pyrone moieties of baicalein and quercetin formed hydrogen bonds with Ser530. The remaining contacts between these herbal compounds (baicalein and quercetin) and the residues lining the hydrophobic active site channel nearby Arg120 in PTGS2 occurred via van der Waals interactions (Figure 7). The results of molecular docking identified 19 interactions between the five tested compounds and potential targets. Among these interactions, PTGS2 and quercetin had the highest binding affinity, followed by PTGS2 and baicalein. In addition, the docking scores of MAPK3, MAPK1, and AKT1 were high when interacting with quercetin. Quercetin may work most effectively with atherosclerosis-related genes because this compound was linked to a lot of atherosclerosis-related genes with high confidence scores in network analysis. And this compound had the highest affinity binding score against six proteins in the molecular docking.

## 3. Discussion

Many herbal formulas are effective for treating complicated diseases involving multiple targets and components as well as disorders that require synergic interactions between compounds for therapeutic benefit [20,32,36]. However, due to the complex ingredient composition of herbal formulas, it is extremely difficult to investigate their role in the body [37]. Thus, the network pharmacology method has been applied in an attempt to acquire a systemic and more complete understanding of the efficacy and target pathways of herbal formulas [6,38]. The herbal formula GBH is frequently used for clinical treatment of cardiovascular disease and contains multiple compounds, some of which exert therapeutic effects on atherosclerosis. Therefore, in the present study, we performed a network pharmacology-based prediction of herb-compound-target and gene-pathway interactions for GBH against atherosclerosis.

Our chemical profiling show that twenty-one compounds were identified in GBH methanol extract using GC-MS and LC-MS. GBH exhibits pharmacological effect for whole active compounds, including not only major active compounds but also minor active compounds. Therefore, we extracted for all compounds in GBH using GC-MS and LC-MS analysis, consulting previous reports citing possible compounds, and in conjunction with public databases using ADME in silico screening. We found 18 compounds in GBH and 213 atherosclerosis-related genes in one huge network (exceptions were β-sitosterol α1 (compound) and MSMO1 (gene)). The main node compounds of GBH were quercetin, kaempferol, baicalein, ellagic acid, and baicalin; these compounds had a high degree of node and betweenness centrality in this network. Xiao et al. (2017) showed that quercetin alleviates atherosclerosis by regulating the expression of nicotinamide adenine dinucleotide phosphate (NADPH) oxidase subunits in vivo and suppresses the overproduction of reactive oxygen species (ROS) stimulated by oxidized low-density lipoprotein (ox-LDL) in mouse peritoneal macrophages [39]. Kaempferol provides benefits in the treatment of atherosclerosis through its antioxidant and anti-inflammatory properties [40], and it alleviates ox-LDL-induced apoptosis by inhibiting the PI3K/Akt/mTOR pathway and increasing autophagy in human endothelial cells [41]. Baicalein is a clinically available anti-atherosclerotic compound that modulates a nitric oxide-related mechanism under oxidized LDL exposure [42] and protects against inflammation by modulating AMPK-α [43]. Ellagic acid has atheroprotective effects on THP-1 monocytes and human umbilical vein endothelial cells and inhibits a cellular adhesion molecule and a pro-inflammatory cytokine (interleukin-6) [44]. Baicalin potentially exerts anti-atherosclerosis effects through the PPARγ–LXRα–ABCA1/ABCG1 pathway by promoting cholesterol efflux from macrophages and delaying the formation of foam cells [45]. In addition to the five node compounds (that is, quercetin, kaempferol, baicalein, ellagic acid, and baicalin), several other compounds present in GBH help to prevent atherosclerosis. Cinnamaldehyde significantly suppresses ox-LDL-induced vascular smooth muscle cell (VSMC) proliferation and migration, inflammatory cytokine overproduction, and foam cell formation in VSMCs and macrophages [30]. Paeoniflorin might alleviate inflammation associated with atherosclerosis by inhibiting the NF-kB pathway [29]. Paeonol inhibits proliferation of VSMC2 by increasing autophagy and activating the AMPK-mTOR signaling pathway [31].

In this network, six genes, AKT1, CASP3, MAPK1, MAPK3, NOS2, and PTGS2, were selected as potential core genes because of a high degree of node and closeness centrality. AKT1 is a major regulator of apoptosis in VSMCs that protects against arterial remodeling and atherosclerosis in vivo [46]. CASP3 is a key executioner protease in the apoptotic pathway that has been identified in atherosclerotic plaques. Grootaert et al. (2016) reported that CASP3 deletion promotes plaque growth and plaque necrosis in ApoE-knockout mice [47]. MAPK1 and MAPK3 are serine/threonine kinases, and the expression of genes encoding these kinases is implicated in macrophage cholesterol homeostasis and atherosclerosis-related gene expression in a cell-based model of atherosclerosis [48]. NOS2 and PTGS2 are inducible enzymes, examples of which include inducible nitric oxide synthase (iNOS) and cyclooxygenase-2 (COX-2). iNOS is present in human atherosclerotic plaques and is involved in the inflammatory process. Previous reports show that iNOS activity is related to the size of atherosclerotic lesions in ApoE-deficient mice [49]. The role of COX-2 in atherosclerosis is complex because its effects are mediated by a variety of eicosanoids with pro- or anti-atherogenic effects that may vary during the evolution of plaques [50].

Our KEGG pathway analysis predicted that these targets are mainly associated with signal transduction-related pathways such as TNF, HIF-1, FoxO, and PI3K-Akt signaling pathways. As a pro-atherogenic cytokine, TNF promotes the expression of cytokines and adhesion molecules, as well as the migration and mitogenesis of VSMCs and endothelial cells [51]. HIF-1α plays a significant role in the responses of endothelial cells, VSMCs, and macrophages, and thereby promotes the development of atherosclerosis [52]. FoxO is one of the downstream signals regulated by activated AKT; endothelial FoxO ablation has atheroprotective effects in low-density lipoprotein receptor knockout mice, and in mouse endothelial cells [53]. The PI3K-Akt signaling pathway affects multiple cell types and pathway cascades related to atherosclerosis [54]. Activation of this pathway through vascular endothelial dysfunction in atherosclerotic animal models reduces the formation of atherosclerotic plaques [55,56].

Our finding that GBH contains five core compounds (quercetin, kaempferol, baicalein, ellagic acid, and baicalin) that can be linked to six potential target genes (AKT1, CASP3, MAPK1, MAPK3, NOS2, and PTGS2) related to atherosclerosis is in agreement with previous reports. These six potential target proteins associated with five active compounds selected by network pharmacology analysis were further analyzed by molecular docking simulation. This determined the binding pose of the active compounds when bound to the target protein, and suggested that the active compounds might have high binding affinity for proteins encoded by atherosclerosis-related genes. The most important node (quercetin), which had a high degree, closeness centrality, and betweenness centrality derived from network analysis, had the highest affinity with potential target genes in the molecular docking score. Also, previous reports revealed that quercetin effectively alleviates atherosclerosis [39]. This result indicates that rather than one compound interacting with one target protein, one or more compounds could interact with multiple target proteins. Thus, we confirmed the hypothesis that the efficacy of the GBH herbal formula is a result of the interaction of many active compounds and multiple target genes and/or proteins.

## 4. Materials and Methods 

The process of network analysis was carried out as follows: (1) Information on the ingredients in GBH was obtained using our chemical profiling results, previous chemical profiling studies, and public herbal medicine databases, and structures and names of the ingredients were confirmed using public chemical databases. (2) Ingredients were filtered using an ADME system. (3) Selected ingredients were associated with human genes using public databases. (4) Sorted genes were linked to the target disease (atherosclerosis). (5) Pathway analysis was carried out using databases [38,57]. (6) Molecular docking analysis was carried out on the core compounds and potential target genes. Details of each process are described below and summarized in Figure 8.

### 4.1. Selection of Compounds in GBH

Herbal ingredients in GBH and their efficacy were searched using the following public databases: OASIS (https://oasis.kiom.re.kr/updated 25 March 2020) and the Traditional Chinese Medicine System Pharmacology Database and Analysis Platform (TCMSP; http://lsp.nwu.edu.cn/, version 2.3) [7,33]. GBH is a mixture of five whole herbal medicines (bark of *Cinnamomum cassia* J. Peresl, sclerotium of *Poria cocos* Wolf, roof of *Paeonia suffruticosa* Andrews, root of *Paeonia lactiflora* Palla, and seed of *Prunus persica* Batsch) rather than a prepared drug extract, and it is classified as a blood-activating and stasis-dispelling herbal formula. We searched for active compounds based on our chemical profiling and previous studies that used High Performance Liquid Chromatography (HPLC) and LC-MS analysis of major active compounds in the GBH formula [10,26]. To identify its components, GBH was extracted in methanol, and we performed chemical profiling using gas chromatography mass spectrometry (GC-MS) and liquid chromatography quadrupole mass detector (LC-MS). A total of 100 mg of GBH was ultrasonicated in 10 mL of methanol for 60 min and filtered through a 0.2 μm syringe filter before GC-MS (5977A series, Agilent Technologies, Santa Clara, CA, USA) and LC-MS (Waters Corporation, Milford, MS, USA) analysis. GC-MS analysis performed as using a DB-5 column (30 m × 0.25 mm id, 0.25 μm thickness, 5% diphenyl, 95% dimethylsiloxane; Agilent Technologies). The oven temperature was set at 50 °C for 2 min, then increased to 300 °C at 10 °C/min and maintained for 10 min. The MS interface and GC injector were maintained at 280 °C and 250 °C, respectively, under 1mL/min for He gas, 70eV for electron energy, 1μL injection volume, and splitless mode. LC-MS analysis was carried out using the Luna C18(2) column (4.6 × 250 mm, 5 μm, Phenomenex, Inc., Torrance, CA, USA). Mobile phase was used with 0.05% formic acid in distilled water (A) and 0.05% formic acid in acetonitrile (B), with the gradient mode from 100% A to 0% A for 60 min. The flow rate was set at 0.8 mL/min, with 10 μL of injection volume and 270 nm of wavelength. A Quadrupole Dalton (QDa) detector was set as follows: nitrogen gas, electrospray ionization (ESI) capillary with positive and negative mode, 0.8kV of ESI capillary, 600 °C as probe temperature, 15 V of con voltage, 120 °C as source temperature, and 20:1 spilt ratio. Additional information about minor active compounds, which might also have pharmacological efficacy, was extracted from the public database for herbal medicines [33,58]. Chemical structures, synonyms, molecular weight, 2D structures, chemical numbers, and physicochemical properties were confirmed using public databases such as Chemspider (http://www.chemspider.com/, version 2020.0.10.0) and PubChem (https://pubchem.ncbi.nlm.nih.gov/, updated 25 March 2020) [32].

### 4.2. Pharmacokinetic ADME Screening

An in silico integrative ADME model was used to select compounds with favorable pharmacokinetic properties using the TCMSP [20,57,58], including oral bioavailability (OB) ≥ 30% and drug likeness (DL) ≥ 0.18, based on criteria suggested by the TCMSP [7]. The oral route is commonly used to administer herbal medicines [59]. OB is one of the most important pharmacokinetic parameters, and a high OB is often a key indicator of the DL value of a bioactive molecule [37]. DL evaluation helps to identify promising drug candidates during the early stages of drug development [37].

### 4.3. Target Genes Related to the Selected Ingredients

Target genes linked to the selected compounds in GBH were searched using STITCH (http://stitch.embl.de/, version 5.0), a database of known and predicted interactions between ingredients and genes [20], with the “*Homo sapiens*” species setting [60]. STITCH combines sources of protein-compound interactions from experimental databases, pathway databases, drug-target databases, text mining, and drug-target predictions into a unified network [61]. Compound-protein interactions with a similarity search value ≥ 0.700, which is a high-confidence score according to STITCH [62], were screened. The confidence score for protein-compound interactions was multiplied with the protein’s expression percentiles in the STITCH database [57]. Gene information including gene ID, name, and organism was confirmed using UniProt (https://www.uniprot.org/, updated in 15 October 2019) [32].

### 4.4. Potential Target Genes and Construction of Networks

Human gene information related to atherosclerosis was extracted from the Therapeutic Target Database (https://db.idrblab.org/ttd/, updated 25 March 2020) and GeneCards: The Human Gene Database (https://www.genecards.org/, version 4.14) [7,63]. Genes linked to atherosclerosis and genes associated with the compounds were selected to identify overlapping genes. Networks of herb-compound, compound-gene, and herb-compound-gene interactions were visualized using Cytoscape version 3.7.2 (http://www.cytoscape.org/) without weights. In the graphical network plot, nodes represent herbs and compounds, and edges represent herb-compound and compound-target gene interactions [34].

### 4.5. Pathway Analyses

Pathway enrichment analyses were performed on the target data using the Database for Annotation, Visualization and Integrated Discovery (DAVID), version 6.8 (https://david.ncifcrf.gov/), and the Kyoto Encyclopedia of Genes and Genomes (KEGG; https://www.genome.jp/kegg/, updated 1 July 2018). Only terms with a *p*-value < 0.05 were selected [38,63].

### 4.6. Molecular Docking Studies

We performed docking studies using Schrödinger Suite 2020-2 (Schrödinger, LLC, New York, NY, USA, 2020) to predict the binding models of GBH compounds (baicalein, baicalin, ellagic acid, kaempferol, and quercetin) against AKT1, CASP3, MAPK1, MAPK3, NOS2, and PTGS2 proteins using the network pharmacology approach. The X-ray crystal structures of target proteins were obtained from the Protein Data Bank (https://www.rcsb.org; AKT1: PDB code 6CCY; CASP3: PDB code 2XYG; MAPK1: PDB code 6SLG; MAPK3: PDB code 4QTB; NOS2: PDB code 3E7G; PTGS2: PDB code 5IKR). Each protein structure was revised using Protein Preparation Wizard in Maestro version 12.4 with default parameters. The receptor grid box for docking was generated to 25 Å × 25 Å × 25 Å dimensions centered on the ligand-protein complex in the binding site for all proteins. The five compounds were minimized using an OPLS3e force field with a dielectric constant of 80.0 in MacroModel version 12.8. The docking studies of compounds were performed using the standard precision method in Glide version 8.7 with default parameters. The predicted binding models of compounds for six target proteins were represented using Discovery Studio 2020 (Dassault Systèmes BIOVIA, San Diego, CA, USA, 2020).

## 5. Conclusions

Our network pharmacology results and subsequent molecular docking analysis not only provide information about the mechanisms that underlie the medicinal effects of GBH in atherosclerosis, but also provide a scientific rationale for the clinical use of GBH in targeting blood stasis associated with atherosclerosis. The compounds, target genes, and pathways identified herein will form the basis for future investigation of the clinical effects of GBH.

## Figures and Tables

**Figure 1 plants-09-01750-f001:**
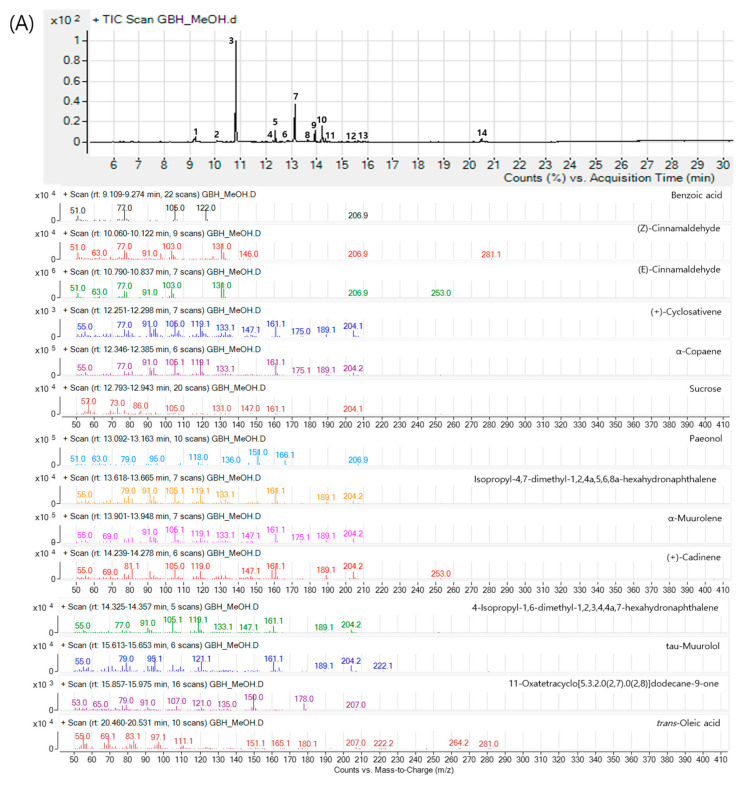
Chemical profiling for Gyejibokryeong-hwan (GBH) using (**A**) gas chromatography mass spectrometry (GC-MS) profiling for (1) benzoic acid, (2) (Z)-cinnamaldehyde, (3) (E)-cinnamaldehyde, (4) (+)-cyclosativene, (5) α-copaene (6) sucrose, (7) paeonol, (8) isopropyl-4,7-dimethyl-1,2,4a,5,6,8a-hexahydronaphthalene, (9) α-muurlene, (10) (+)-cadinene, (11) 4-isopropyl-1,6-dimethyl-1,2,3,4,4a,7-hexahydronaphthalene, (12) tau-muurolol, (13) 11-oxatetracyclo[5.3.2.0(2,7).0(2,8)]dodecane-9-one, and (14) trans-oleic acid and (**B**) liquid chromatography quadrupole dalton mass detector (LC-MS) for (1) amygdalin, (2) paeonolide, (3) albiflorin, (4) paeoniflorin, (5) ellagic acid, (6) benzoylpaeoniflorin, (7) trans-cinnamic acid, (8) paeonol, and (9) (E)-cinnamaldehyde.

**Figure 2 plants-09-01750-f002:**
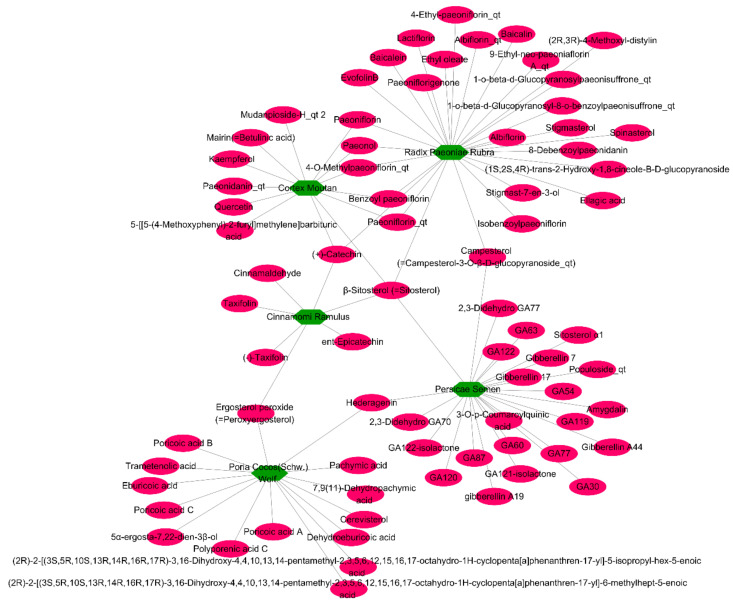
A Network of five herbal medicines (green hexagons) and the 74 final compounds (pink ovals) selected through absorption, distribution, metabolism, and excretion (ADME).

**Figure 3 plants-09-01750-f003:**
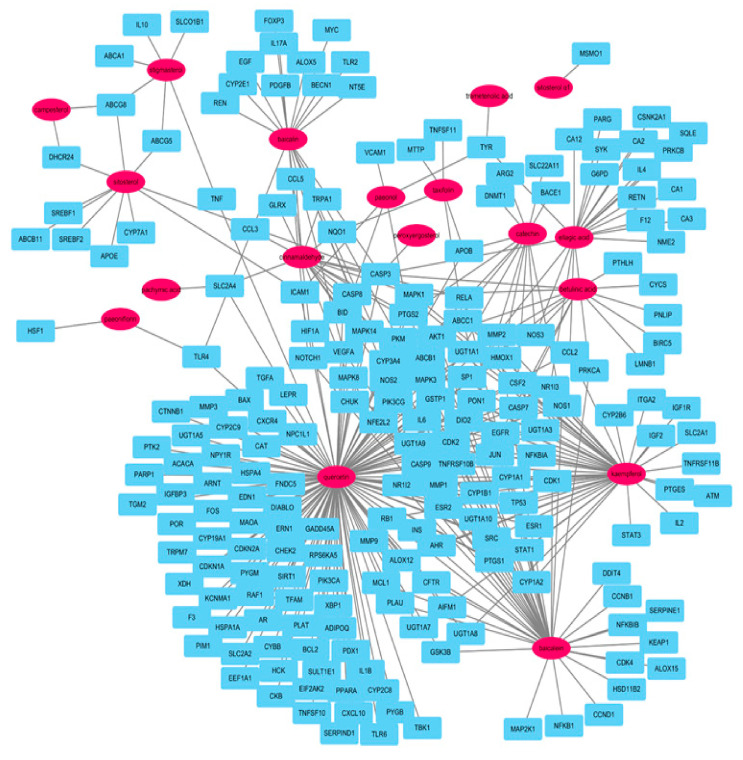
Correlation between compounds (pink ovals) and atherosclerosis-related genes (cyan rectangles) in GBH.

**Figure 4 plants-09-01750-f004:**
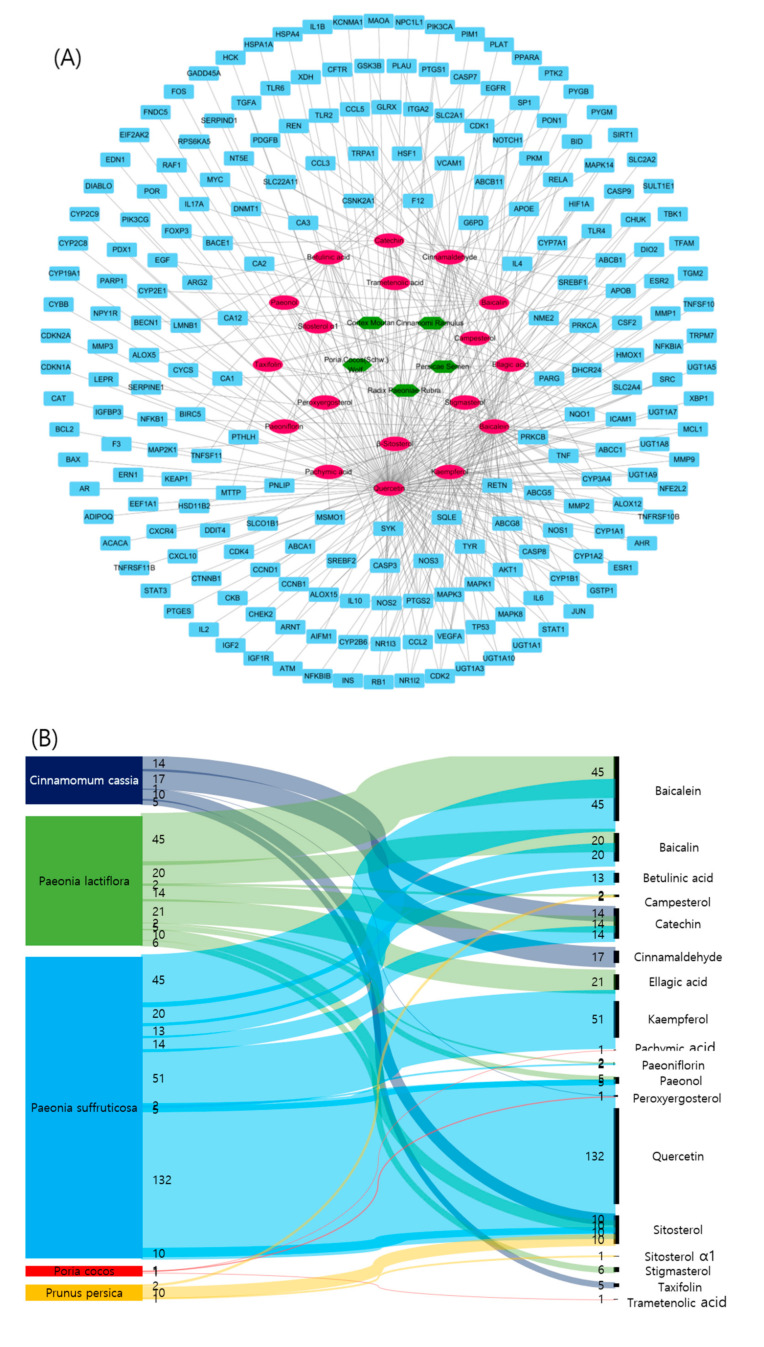
Network and Sankey diagrams of herbs-compounds-genes for GBH. (**A**) Network of herbs-compounds-genes for GBH; herbs are indicated in green hexagons, compounds are indicated in pink ovals, and target genes are indicated in cyan rectangles. (**B**) A Sankey diagram of herbs and active compounds containing atherosclerosis-related genes.

**Figure 5 plants-09-01750-f005:**
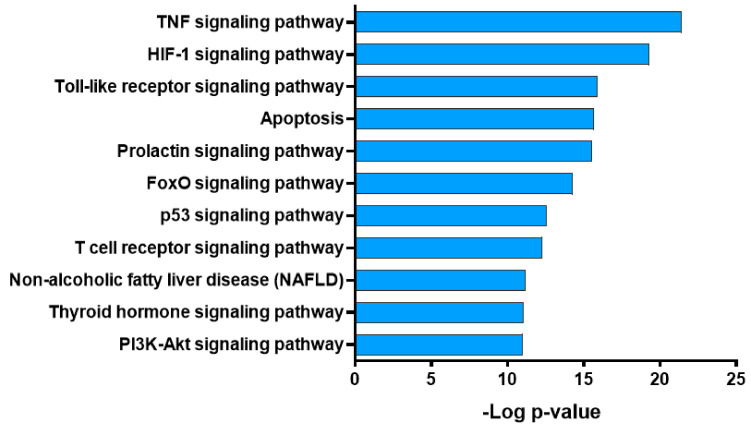
Kyoto Encyclopedia of Genes and Genomes (KEGG) pathway enrichment analysis of potential target genes of Gyejibokryeong-hwan (GBH).

**Figure 6 plants-09-01750-f006:**
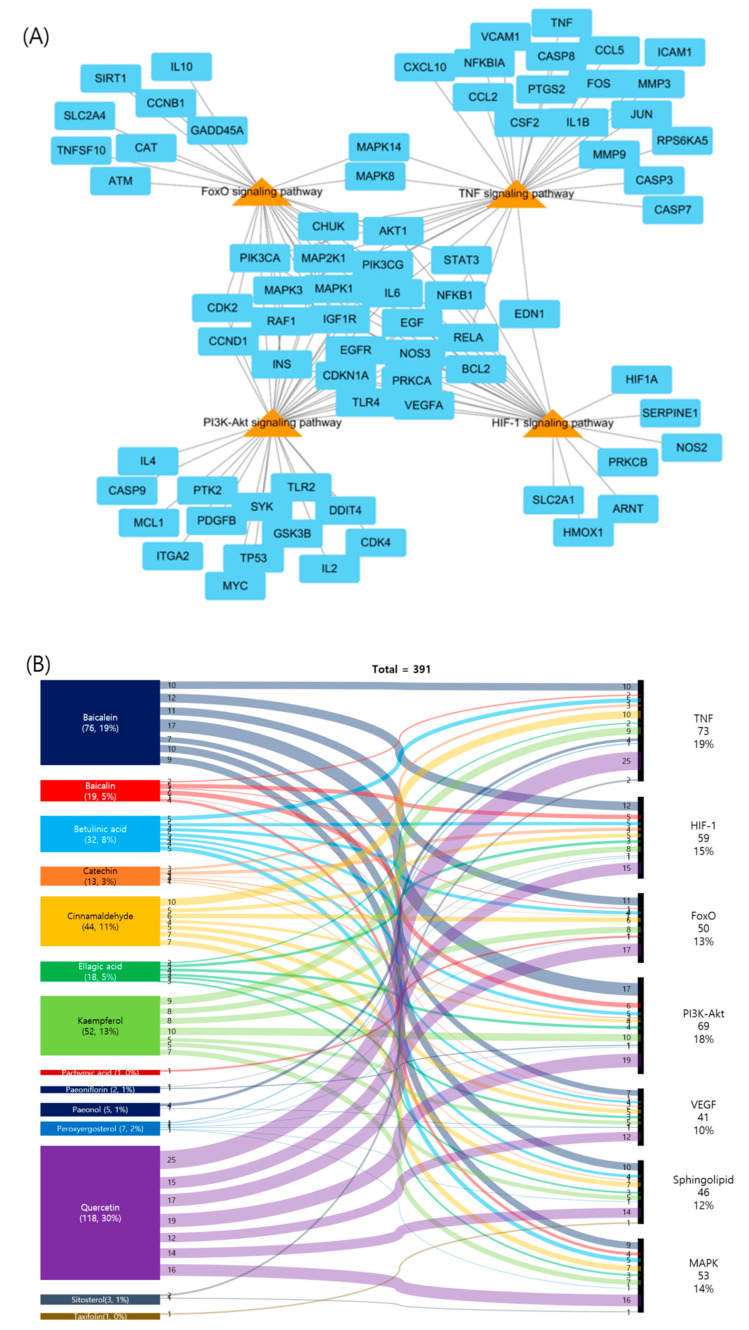
Network and Sankey diagrams of target gene–KEGG pathway for Gyejibokryeong-hwan (GBH). (**A**) Target gene–KEGG pathway network for GBH; genes are represented by cyan rectangles, and pathways are indicated by orange triangles. (**B**) A Sankey diagram of compounds acting on the KEGG pathways.

**Figure 7 plants-09-01750-f007:**
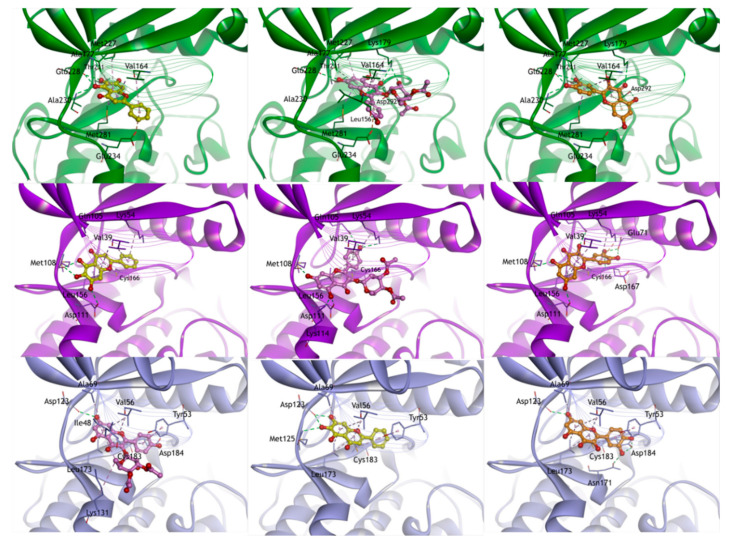
Docking models of GBH analogues (baicalein, baicalin, ellagic acid, kaempferol, and quercetin) to potential target proteins (AKT1, CASP3, MAPK1, MAPK3, NS2, and PTGS2). Note: Compounds are represented by ball and stick models, in which the colors are yellow for baicalein, yellow-green for baicalin, cyan for ellagic acid, pink for kaempferol, and orange for quercetin. All proteins are shown in ribbon model, and the colors are green for AKT1, red for CASP3, purple for MAPK1, lavender for MAPK3, blue for NOS, and pink for PTGS2. For clarity, key binding site residues are shown in sticks and labeled using the three-letter amino acid code. The hydrogen bonds are displayed as green dashed lines, and hydrophobic interactions are shown as pink dashed lines.

**Figure 8 plants-09-01750-f008:**
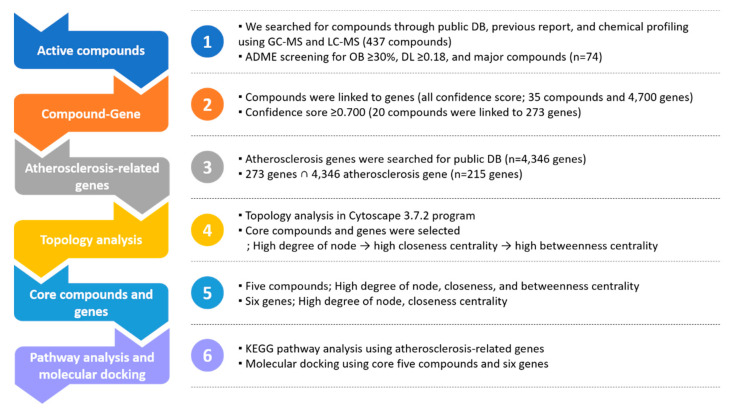
Workflow of network pharmacological analysis of Gyejibokryeong-hwan (GBH).

**Table 1 plants-09-01750-t001:** Compounds selected through screening (oral bioavailability ≥ 30% and drug-likeness ≥ 0.18) linked to target genes with a confidence score ≥ 0.700.

Components	Genes	Degree	Closeness Centrality	Betweenness Centrality
Amygdaline	KLRP	1	1	0
Baicalein	ABCC1, AKT1, ALOX12, ALOX15, CCNB1, CCND1, CDK1, CDK2, CDK4, CFTR, CYP1A1, CYP1A2, CYP1B1, CYP3A4, DDIT4, GSK3B, HSD11B2, IL6, INS, KEAP1, MAP2K1, MAPK1, MAPK3, MAPK8, MCL1, MITF, MMP2, MMP9, NFE2L2, NFKB1, NFKBIB, NOS1, NOS2, NOTCH1, NR1I2, PKM, PLAU, PRKCA, PTGS1, PTGS2, RB1, RELA, S100A7, SERPINE1, TNFRSF10B, TP53, UGT2B15, VEGFA	48	0.36213	0.13466
Baicalin	ALOX5, BECN1, CASP3, CASP8, CASP10, CDX1, CDX2, CYP2E1, CYP3A4, EGF, FOXP3, HIF1A, IL17A, MYC, NOS2, NOTCH1, NT5E, PDGFB, PKM, REN, TLR2, TLR4, VEGFA	23	0.34969	0.09976
Betulinic acid	AKT1, BIRC5, CASP3, CASP7, CYCS, EGFR, LMNB1, MAPK1, MAPK3, NOS3, PNLIP, PTHLH, SP1, TOP1, TOP2A	15	0.33969	0.05121
Campesterol	ABCG8, DHCR24, HSD3B2	3	0.2	0.00708
Catechin	ABCB1, APOB, ARG2, BACE1, CSF2, DNMT1, HMOX1, IL6, KIAA1149, NOS1, NOS2, NOS3, PON1, PTGS2, SLC22A11, SLC35A2, SLC47A1, ZFP36	18	0.33648	0.05808
Cinnamaldehyde	AKR1C2, AKT1, BID, CASP3, CASP8, CCL3, CCL5, GRKL, GLRX, IL8, MAPK1, MAPK3, MAPK8, MAPK14, NOS2, NQO1, PTGS2, RELA, SLC2A4, TRPA1	20	0.35404	0.05899
Dehydroeburicoic acid	SPTAN1	1	1	0
Ellagic acid	CA1, CA2, CA3, CA4, CA5A, CA5B, CA6, CA7, CA9, CA12, CA14, CASP3, CCL2, CSNK2A1, F12, G6PD, IL4, MMP2, NME1-NME2, NME2, NOS3, NR1I3, PARG, PGD, PRKACA, PRKCA, PRKCB, RETN, SQLE, SYK, TMPRSS11D, TYR	32	0.35670	0.17266
Kaempferol	ABCB1, ABCC1, AHR, AKT1, ALOX12, ATM, CASP3, CASP9, CCL2, CDK1, CDK2, CHUK, CSF2, CYP1A1, CYP1A2, CYP1B1, CYP2B6, CYP3A4, DIO2, ESR1, ESR2, GSTP1, H2AFX, HMOX1, IGF1R, IGF2, IL2, ITGA2, JUN, MAPK1, MAPK3, MMP1, MMP2, NFKBIA, NOS1, NOS2, NR1I2, NR1I3, PTGES, RB1, RPS6KA3, SLC2A1, SRC, STAT1, STAT3, TNFRSF11B, TP53, UGT1A1, UGT1A3, UGT1A7, UGT1A8, UGT1A9, UGT1A10, UGT3A1	54	0.37748	0.12492
Pachymic acid	SLC2A4	1	0.26611	0
Paeoniflorin	HSF1, IL8, TLR4	3	0.28079	0.00720
Paeonol	CASP8, CASP10, ICAM1, PTGS2, TYR, VCAM1	6	0.31844	0.01390
Peroxyergosterol	CASP3	1	0.31148	0
Quercetin	ABCB1, ABCC1, ABCC4, ABCC5, ACACA, ADIPOQ, AHR, AIFM1, AKR1C3, AKT1, ALOX12, AOX1, APAF1, APOB, AR, ARNT, ATP5A1, ATP5B, ATP5C1, BAX, BCL2, BID, CASP3, CASP7, CASP8, CASP9, CAT, CCL2, CD97, CDK2, CDKN1A, CDKN2A, CFTR, CHEK2, CHUK, CKB, CSF2, CTNNB1, CXCL10, CXCR4, CYBB, CYP1A1, CYP1A2, CYP1B1, CYP2C8, CYP2C9, CYP2D6, CYP3A4, CYP19A1, DIABLO, DIO2, EDN1, EEF1A1, EGFR, EIF2AK2, ERN1, ERN2, ESR1, ESR2, F3, FAU, FNDC5, FOS, FOXM1, GADD45A, GLRA1, GSK3B, GSTP1, HCK, HIBCH, HIF1A, HIST3H3, HMOX1, HSPA1A, HSPA4, ICAM1, IGFBP3, IL1B, IL6, IL8, JUN, KCNMA1, KRAS, LEPR, MAOA, MAPK1, MAPK3, MAPK8, MAPK14, MCL1, MMP1, MMP2, MMP3, MMP9, NFE2L2, NFKBIA, NOS1, NOS2, NOS3, NPC1L1, NPY1R, NR1I2, NR1I3, PARP1, PDX1, PIK3CA, PIK3CG, PIM1, PLAT, PLAU, PON1, POR, PPARA, PTGS1, PTGS2, PTK2, PYGB, PYGL, PYGM, RAF1, RB1, RNASEL, RPS6KA5, SERPIND1, SIRT1, SLC2A2, SLC2A4, SP1, SRC, STAT1, STK17B, SULT1A1, SULT1E1, TBK1, TFAM, TGFA, TGM2, TLR4, TLR6, TMPRSS11D, TNF, TNFRSF10B, TNFSF10, TP53, TRPM7, UGT1A1, UGT1A3, UGT1A5, UGT1A7, UGT1A8, UGT1A9, UGT1A10, UGT2A3, UGT2B15, UGT2B4, UGT3A1, VEGFA, XBP1, XDH	161	0.54493	0.72132
Stigmasterol	ABCA1, ABCG5, ABCG8, IL8, IL10, SLCO1B1, TNF	7	0.27859	0.02904
Sitosterol	ABCB11, ABCG5, ABCG8, APOE, CASP3, CYP7A1, DHCR24, ICAM1, SREBF1, SREBF2	10	0.32350	0.06171
Sitosterol alpha1	C5orf4, MSMO1	2	1	1
Taxifolin	ABCC1, APOB, MTTP, NQO1, TNFSF11	5	0.29171	0.01483
Trametenolic acid	TYR	1	0.21064	0

**Table 2 plants-09-01750-t002:** Docking scores of the active compounds of GBH with their potential targets.

Targets	PDB Code	Compounds	Docking Score
AKT1	6CCY	Baicalein	−6.51
AKT1	6CCY	Kaempferol	−5.54
AKT1	6CCY	Quercetin	−7.28
CASP3	2XYG	Baicalin	−6.07
CASP3	2XYG	Ellagic acid	−5.98
CASP3	2XYG	Kaempferol	−5.21
CASP3	2XYG	Quercetin	−6.21
MAPK1	6SLG	Baicalein	−6.55
MAPK1	6SLG	Kaempferol	−6.71
MAPK1	6SLG	Quercetin	−7.73
MAPK3	4QTB	Baicalein	−7.01
MAPK3	4QTB	Kaempferol	−6.04
MAPK3	4QTB	Quercetin	−7.76
NOS2	3E7G	Baicalein	−5.70
NOS2	3E7G	Baicalin	−6.10
NOS2	3E7G	Kaempferol	−3.44
NOS2	3R7G	Quercetin	−7.13
PTGS2	5IKR	Baicalein	−8.72
PTGS2	5IKR	Quercetin	−9.61

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
