# Peer review of "Exploring the Mechanism of Gyejibokryeong-hwan against Atherosclerosis Using Network Pharmacology and Molecular Docking"

_plants, 2020, doi:10.3390/plants9121750_

Round 1
Reviewer 1 Report
Dear Authors;
In this study you analyzed based on network pharmacology approach and molecular docking simulation the effect and mechanism of Gyejibokryeong‑hwan (GBH), an herbal medicine widely used in the treatment of atherosclerosis.
You claim that this formulation is traditional used to treat atherosclerosis, however you barely mention; “protection against atherosclerosis” you must provide a strong background, of the traditional use of the preparation to treat the disease, how use it, how it is prepared, doses, which plants parts are used, etc.
The correct botanical names of the plant are mandatory.
How do you know that each reported compound for a plant is present in the traditional preparation? the presence or absence of a compound will depend on the solvent, used part and preparation, so you must perform a phytochemical analysis of the traditional used preparation, then you can assume the presence of the selected compounds.
Author Response
Dear, editor,
We appreciate the opportunity to respond to our reviewer’s comments. Our manuscript is now revised to incorporate reviewer’s insightful suggestions and critiques. We have modified the 3 Figures, and added 2 reviewer’s Figure and Supplementary Table 1 to address the reviewers’ concerns. Also, our manuscript has been revised and supplemented by English editing by Bioedit corporation.
(Online address of editing certificate [https://www.bioedit.com/digital-certificate/view/dd9b4233f63b7893417b3e2e503b8e6fe8905bc2])
Major changes in the text are highlighted in red (edited in English) and blue (edited in response to reviewer comments) in the revised manuscript. In addition, below please find a point-by-point response to the reviewer’s comments.

Reviewer 2 Report
The manuscript by Lee et et al investigates the effects and mechanism of the herbal medicine Gyejibokryeong-hwan against atherosclerosis by network pharmacology and molecular docking calculations. Although the topic is of interest I found several flaws in the manuscript which, in addition, requires extensive English corrections. The part concerning molecular docking calculations is not convincing and needs to be further developed.
- Introduction has to be improved: When explaining the role of molecular docking calculations significant references should be included
- The paragraph 6. Molecular docking verification has to be rewritten and not only for the English: It is not clear how the authors could evaluate “low binding energy” the scores generated by the docking program; please explain the phrase: “The hydroxy group of the aromatic ring in flavonoids can form hydrogen bonds through hydrophobic interactions with the hinge region of the genes.” ; Docking modes in Figure 7 cannot be analysed because pictures are too small;
- Line 230: Why docking calculations validate the hypothesis that identified compounds and targets are closely related to pathways associated with atherosclerosis? Because of a negative docking score?
- Docking methodology lacks of a proper description and validation
Author Response
Dear, Editor,
We appreciate the opportunity to respond to our reviewer’s comments. Our manuscript is now revised to incorporate reviewer’s insightful suggestions and critiques. We have modified the 3 Figures, and added 2 reviewer’s Figure and Supplementary Table 1 to address the reviewers’ concerns. Also, our manuscript has been revised and supplemented by English editing by Bioedit corporation.
(Online address of editing certificate [https://www.bioedit.com/digital-certificate/view/dd9b4233f63b7893417b3e2e503b8e6fe8905bc2])
Major changes in the text are highlighted in red (edited in English) and blue (edited in response to reviewer comments) in the revised manuscript. In addition, below please find a point-by-point response to the reviewer’s comments.

Reviewer 3 Report
This paper Exploring the Mechanism of Gyejibokryeong-hwan Against Atherosclerosis Using Network Pharmacology and Molecular Docking Approach aims to explain the anti-atherosclerotic pharmacodynamic effect by elucidating the mechanism of action of the natural compounds from Gyejibokryeong-hwan with computational tools, i.e., network pharmacology and molecular docking. The paper is generally interesting; however, it is plagued by several serious conceptual and methodological flaws.
In the Introduction, the authors state that "Unlike western medicine, numerous ingredients in herbal medicine can act on several diverse pathways, including lipid metabolism...", thus putting in opposition the western medicine and herbal medicine. The phrase mentioned earlier is profoundly incorrect because western medicine—evidence-based—also integrates herbal medicine into the scientific method. Phytochemistry and pharmacognosy are well-established pharmaceutical sciences that deal with drugs of natural origin (i.e., medicinal plants) and render valuable outcomes for physical-chemical and pharmacologic characterization, chemical structure elucidation, as well as for drug development. The lines 63-64 confuse the English-language readership, like—unfortunately—many other lines in this manuscript. It is not clear whether the effects on atherosclerosis were reported (or not) in the literature.
After carefully reading this manuscript, the general impression is that of misusing the concept of network pharmacology. Indeed, the complex network analysis tools enable the visualization and the qualitative and quantitative analysis of the graph. However, the most important advantage of using complex network analysis is its predictive power. By building a complex network from known data, one can infer new information/knowledge. It is unclear what new knowledge the authors uncover from the network in Figure 4 (Results, section 2.4 Network analysis of potential compounds and target genes). Is was previously unknown (i.e., before the present paper) that quercetin interacts with 132 atherosclerosis-related genes or that kaempferol interacts with 51 atherosclerosis-related genes? Do the authors predict that the five natural compounds interact with different genes than those listed in GeneCards? Unfortunately, these questions remain unanswered.
Table's 1 caption (in the Results section) should be more comprehensive in that it clarifies what the "combined score" (and what does it represent) is; the authors should also describe or define this score in the main text. Furthermore, the authors should define the network centralities—degree, closeness, and betweenness—and explain their relevance in the network so that the Plants Journal's readership better understand the network procedure and the results.
In subsection 4.4. Potential target genes and construction of networks: the authors should clearly explain how they set the links between nodes within their tripartite graph. Are the links oriented or not? Do the links have weights or not? If the links have weights, then they have to explain their meaning.
In conclusion, although the manuscript has its merits, it does not employ network pharmacology—as claimed—to explore the anti-atherosclerotic mechanism of the five main compounds within GBH. The authors use networks only to visualize the relationships between the plants from GBH—natural compounds—atherosclerosis-related genes. Although they compute network centralities, it is not clear how they use them. The entire paper seems to be plagued by circular logic: the authors use available knowledge to find things that are also already known. They formulate no hypothesis and predict nothing.
Author Response

(The authors gave the same response as above.)

Round 2
Reviewer 1 Report
Dear Authors;
The plant names still are not correct:
As example; Prunus persica Batsch is Prunus persica (L.) Batsch (Rosaceae) at least the correct plant name needs to be mentioned one time.
“In addition, we performed LC-MS and GC-MS pattern analysis for GBH methanol extracts.”
Please integrate the above-mentioned results in the manuscript.
Author Response
7th December, 2020
Professor Dilantha Fernando, Editor-in-Chief,
Plants
Ref. No.: plants-972459
Dear Editor,
We appreciate the second opportunity to respond to our reviewer’s comments. Our manuscript is now revised to incorporate reviewer’s meaningful critiques. We have modified two Figures (add Figure 1 and change Figure 8), and added Supplementary Table 5 to address the reviewers’ concerns. Major changes in the text are highlighted in red in the revised manuscript. In addition, please find below a point-by-point response to the reviewers’ comments.
Response to Reviewer 1 Comments:
Point 1: The plant names still are not correct:
As example; Prunus persica Batsch is Prunus persica (L.) Batsch (Rosaceae) at least the correct plant name needs to be mentioned one time.
Response 1: We thank the reviewer for his/her valuable comment. We have corrected botanical name of the plants: Cinnamomum cassia → Cinnamomum cassia J. Presl (Lauraceae); Poria cocos → Poria cocos Wolf (Polyporaceae); Paeonia suffruticosa → Paeonia suffruticosa Andrews (Paeoniaceae); Paeonia lactiflora → Paeonia lactiflora Pallas (Paeoniaceae); Prunus persica → Prunus persica Batsch (Rosaceae) [lines 46–47, 110–111, 171].
Point 2: “In addition, we performed LC-MS and GC-MS pattern analysis for GBH methanol extracts.” Please integrate the above-mentioned results in the manuscript.
Response 2: We thank the reviewer for his/her valuable comment. We integrated the above-mentioned results section 2.2, Figure 2, and Materials and Methods section 4.2 [lines 78-89, 93-104, 342-356]. As Figure 1 was added, the number of existing Figures were increased by one.
Thank you again for your reconsideration of our work and we hope you find the revised manuscript suitable for publication in Plants.
Best Regards,
Jin Mi Chun
Reviewer 2 Report
The authors have satisfactorily responded to my questionsand made the necessary changes to the manuscript.
Author Response
Dear Editor,
We appreciate the second opportunity to respond to our reviewer’s comments. Our manuscript is now revised to incorporate reviewer’s meaningful critiques. We have modified two Figures (add Figure 1 and change Figure 8), and added Supplementary Table 5 to address the reviewers’ concerns. Major changes in the text are highlighted in red in the revised manuscript. In addition, please find below a point-by-point response to the reviewers’ comments.
Response to Reviewer 2 Comments:
The authors have satisfactorily responded to my questions and made the necessary changes to the manuscript.
Response 1: Thank you for his/her valuable comment from the reviewer.
Thank you again for your reconsideration of our work and we hope you find the revised manuscript suitable for publication in Plants.
Best Regards,
Jin Mi Chun
Reviewer 3 Report
Please, see the attached file.

Author Response
Plants
Ref. No.: plants-972459
Dear Editor,
We appreciate the second opportunity to respond to our reviewer’s comments. Our manuscript is now revised to incorporate reviewer’s meaningful critiques. We have modified two Figures (add Figure 1 and change Figure 8), and added Supplementary Table 5 to address the reviewers’ concerns. Major changes in the text are highlighted in red in the revised manuscript. In addition, please find below a point-by-point response to the reviewers’ comments.
Thank you again for your reconsideration of our work and we hope you find the revised manuscript suitable for publication in Plants.
Best Regards,
Jin Mi Chun
